# Fine-tuning Pocket-Aware Diffusion Models via Denoising Policy Optimization

## Abstract

Structure-based drug design has been accelerated by pocket-aware 3D generative models, yet most methods primarily fit the training distribution and may fall short of satisfying multiple properties required in real-world therapeutic drug discovery. Recently, increasing attention has focused on structure-based molecule optimization (SBMO), which targets fine-grained control over multiple specified molecular properties. In this paper, we present DEPPA, a novel SBMO approach building upon Denoising Diffusion Policy Optimization for fine-tuning a pre-trained pocket-aware diffusion model via reinforcement learning. DEPPA enables optimization over multiple properties, including binding affinity, drug-likeness, synthesizability and diversity. We formulate the reverse denoising process of the pretrained pocket-aware diffusion model as a multi-step Markov Decision Process, where the desired properties that serve as reward signals are evaluated on the final generated ligand molecules. DEPPA incorporates a coarse denoising scheduler during the RL fine-tuning to achieve efficient and effective molecule optimization. Experimental results on the CrossDocked2020 benchmark demonstrate that DEPPA outperforms baselines in binding affinity (Vina Score -8.5 kcal/mol), drug-likeness and diversity while exhibiting competitive performance in synthesizability. The source code is available at
`https://anonymous.4open.science/r/DePPA-5E76`.

## 1 Introduction

Structure-based drug design (SBDD) aims at identifying 3D structure of novel molecules that bind to a specific protein pocket with high affinity (Anderson, 2003; Isert et al., 2023). It has been considered as a crucial and challenging task in drug discovery, due to the enormous chemical space and complex interactions between ligands and protein pockets. With the remarkable advancement in deep generative models for molecules, the efficiency of SBDD has been significantly enhanced by directly generating promising drug candidates for the given protein targets. Specifically, autoregressive models have been proposed for SBDD tasks, following an atom-based (Luo et al., 2021; Peng et al., 2022; Liu et al., 2022a) or fragment-based (Zhang et al., 2023; Powers et al., 2022; Zhang & Liu, 2023) sequential generation process. Given that autoregressive methods tend to suffer from error accumulation, Guan et al. (2023); Schneuing et al. (2024); Huang et al. (2024); Guan et al. (2024) propose non-autoregressive methods based on diffusion models, which target full-molecule generation in an iterative denoising manner.

Despite the promising progress, these generative methods mainly focus on conditional sampling that mimics the underlying training data distribution of ligand-pocket complexes via likelihood maximization, without explicit emphasis on prioritizing multiple desired properties. The effectiveness of a typical generative SBDD method is often limited when the training data does not align well with the desired properties for the task at hand. In such a case, the distribution of task-desired ligand molecules may diverge from the training data distribution, necessitating additional post hoc optimization of the generated ligands—a process that is often resource-intensive and time-consuming.

To address this limitation, increasing attention has recently been directed towards structure-based molecule optimization (SBMO), a more advanced regime within the general scope of SBDD. SBMO aims at generating

potential ligand candidates with high binding affinity while simultaneously ensuring desirable molecular properties such as drug-likeness, synthesizability and diversity. This end-to-end multi-property optimization formulation is essential for meeting the practical demands of real-world therapeutic drug discovery.

Studies in SBMO have explored a variety of optimization paradigms, mostly requiring a sufficiently capable pretrained model to serve as the initial prior. DecompOpt (Zhou et al., 2024) adopts evolutionary-style optimization, while MolJO (Qiu et al., 2024) and TAGMol (Dorna et al., 2024) utilize gradient-based optimization methods. Moreover, it is natural to frame SBMO as a problem of policy optimization. RGA (Fu et al., 2022) employs reinforcement learning to guide the crossover and mutation operations of the evolutionary process. Inspired by language models, 3DMolFormer (Hu et al., 2025a) and BindGPT (Zholus et al., 2025) employ reinforcement learning to guide the autoregressive generation of sequences concatenating SMILES tokens and 3D coordinates, conditioned on the protein pocket as the prompt. However, due to their autoregressive nature, they remain susceptible to error accumulation and can struggle to capture the global features of 3D ligands. ALIDIFF (Gu et al., 2024) and DecompDPO (Cheng et al., 2025) apply direct preference optimization (DPO) (Rafailov et al., 2023; Wallace et al., 2024) to align the pretrained model directly using pairwise preference datasets, which requires task-specific strategies for collecting high-quality preference pairs. While these methods eliminate the cost of reward evaluation in the traditional RL training loop, their DPO objectives are defined over the fixed preference datasets, resulting in a bounded optimization frontier.

**Our Contribution.** In structure-based molecule optimization (SBMO), target properties such as binding affinity, drug-likeness, synthesizability, and molecular diversity are typically available as continuous-valued signals computed by external oracles. Despite their prevalence, the systematic use of these continuous evaluations as reward signals for *fine-tuning pocket-aware diffusion models via online reinforcement learning (RL)* remains largely underexplored in the SBMO literature.

We introduce **De**noising **P**olicy for **P**ocket-**A**ware Molecule Optimization (DEPPA), an SBMO approach that jointly targets multiple molecular properties while ensuring strong binding affinity. By directly integrating the oracles' feedback into the RL-driven fine-tuning process of a pretrained diffusion model, DEPPA enables effective and efficient multi-objective optimization and achieves consistent improvements over the baselines, advancing the Pareto frontier of SBMO.

Inspired by Denoising Diffusion Policy Optimization (DDPO) (Black et al., 2023), we propose to formulate the reverse denoising process of a pre-trained, pocket-conditional diffusion model as a *multi-step Markov Decision Process (MDP)* and apply policy optimization to fine-tune the diffusion model. In this formulation, noisy ligands at different timesteps are treated as *states*, while each denoising step corresponds to an *action*. Binding affinity and the desired molecular properties, which serve as reward signals, are evaluated *only at the final step* of the denoising trajectory on the generated ligand molecules.

Directly applying standard DDPO over the full denoising horizon with sparse, outcome-based rewards can be unstable and computationally expensive, due to error accumulation and the difficulty in long-horizon credit assignment (Hu et al., 2025b). To address these challenges while targeting multiple objectives in the SBMO tasks, we incorporate four key components that are absent from DDPO to improve the efficiency and stability of denoising policy optimization, including critic-free advantage estimation, a coarser denoising schedule, Gaussian rank transformation of rewards, and prediction thresholding. In combination, these key components enable stable and efficient RL fine-tuning of the pretrained pocket-aware diffusion model, without introducing additional parameters to train. Remarkably, by following the variational diffusion framework (Kingma et al., 2023) employed by DiffSBDD (Schneuing et al., 2024), DEPPA establishes a principled connection between the denoising step size and the extent of the denoising policy's exploration.

Experimental results on the CrossDocked2020 benchmark (Francoeur et al., 2020) demonstrate that our approach significantly outperforms state-of-the-art baselines in binding affinity while achieving the best or highly competitive performance in drug-likeness, synthesizability and diversity. To the best of our knowledge, DEPPA is the first to achieve a Vina Score of -8.5 kcal/mol – representing a 33.7% improvement over the reference molecules – while maintaining an average molecular size comparable to the reference level. Moreover, we introduce a variant of DEPPA that applies a top-$N$ selection scheme to all generated ligands collected throughout the complete fine-tuning process. This variant achieves an even more pronounced

advantage in binding affinity, reaching Vina Score of -13.52 kcal/mol, with moderate decreases in molecular properties. In addition, we evaluate the conformational stability of the generated poses. Our approach achieves the best performance in binding-mode consistency, as measured by RMSD, while also exhibiting stronger performance in terms of strain energy and steric clashes.

## 2 Related Work

**Structure-based Drug Design** Structure-based drug design (SBDD) involves generating 3D ligand molecules that exhibit high binding affinity to the target protein pocket while satisfying desirable pharmacological properties. In recent years, a spectrum of deep generative methods have been proposed for SBDD tasks. liGAN (Ragoza et al., 2022) employs a conditional variational autoencoder to generate ligands in atomic density grids. Luo et al. (2021); Peng et al. (2022); Liu et al. (2022a) propose atom-based autoregressive models, while Zhang et al. (2023); Powers et al. (2022); Zhang & Liu (2023) adopt fragment-based autoregressive methods to generate ligands with more realistic substructures. Autoregressive methods tend to suffer from error accumulation and struggle to effectively capture the global structure of 3D molecules. To address the limitations, non-autoregressive methods based on diffusion models or Bayesian flow networks (Guan et al., 2023; Schneuing et al., 2024; Huang et al., 2024; Guan et al., 2024; Qu et al., 2024) are proposed to achieve full-molecule generation, achieving state-of-the-art performance. These diffusion-based methods employ equivariant networks to ensure equivariant likelihood modeling with respect to SE(3) transformations.

**Structure-based Molecule Optimization** Essentially, a generative SBDD method achieves pocket-conditional sampling that mimics the underlying training data distribution of ligand-pocket complexes via likelihood maximization. When the properties targeted by the task are underrepresented in the training data, generation quality may be compromised due to a potential mismatch between the desired molecule distribution and the training data distribution. This limitation poses the need for structure-based molecular optimization (SBMO), which involves explicit fine-grained control over multiple properties of the generated ligands, while ensuring high binding affinity to the target pocket. DecompOpt (Zhou et al., 2024) adopts evolutionary algorithms to approach SBMO while MolJO (Qiu et al., 2024) and TAGMol (Dorna et al., 2024) leverage gradient-based optimization methods. RL-based SBMO has also been explored in recent studies. RGA (Fu et al., 2022) employs reinforcement learning to guide the mutation and crossover operations for molecule evolution. 3DMolFormer (Hu et al., 2025a) and BindGPT (Zholus et al., 2025) adopt token generation models as RL agents to autoregressively generate concatenated sequences of SMILES tokens and 3D coordinates conditioned on the protein pocket as the prompt. Building upon direct preference optimization (DPO) (Rafailov et al., 2023; Wallace et al., 2024), ALIDIFF (Gu et al., 2024) fine-tunes a pretrained diffusion model to achieve explicit binding affinity optimization, while DecompDPO (Cheng et al., 2025) advances this paradigm by accounting for multi-granularity preferences. In contrast to ALIDIFF and DecompDPO, which are trained on a fixed, pre-collected preference dataset, DEPPA operates under the online RL setup, reflecting a fundamental distinction in supervision setting. Concurrent with our work, SeFMol (Zhang et al., 2026) fine-tunes a pretrained diffusion model conditioned on both molecular properties and protein pockets, using a PPO-style update combined with a KL constraint relative to the pretrained model. In contrast, we propose to fine-tune a pretrained diffusion model conditioned solely on protein pockets for multi-objective molecule optimization, using critic-free policy optimization without the KL constraint.

**Controllable Diffusion Models** Diffusion models (Sohl-Dickstein et al., 2015; Ho et al., 2020; Song & Ermon, 2020) have recently achieved remarkable success in generative modeling across various domains. Particularly, EDM (Hoogeboom et al., 2022) manages to incorporate a variant of equivariant graph neural networks (Satorras et al., 2021) into diffusion models to achieve efficient distribution modeling of 3D molecular structures. In order to acquire fine-grained controllability of the diffusion models, recent research in image domain has explored various directions, such as augmenting the embedding space (Gal et al., 2022), compositional diffusion generation (Liu et al., 2022b; Du et al., 2023), classifier guidance (Dhariwal & Nichol, 2021) and classifier-free guidance (Ho & Salimans, 2022). Moreover, Black et al. (2023); Fan et al. (2023) consider the denoising process as a multi-step Markov Decision Process (MDP) and apply reinforcement learning to fine-tune pretrained models. Wallace et al. (2024); Yang et al. (2024) apply direct preference optimization (Rafailov et al., 2023) to fine-tune the pretrained diffusion models directly from pairwise preference data. More recently, diffusion-based SBMO methods have explored the application of classifier guidance

(Dorna et al., 2024) and direct preference optimization (Gu et al., 2024). However, online reinforcement learning–based fine-tuning of diffusion models for SBMO remains largely underexplored.

## 3 Preliminaries

### 3.1 Notations and Problem Definition

A protein binding site is characterized in form of an atomic 3D point cloud $\mathcal{P} = \{(\mathbf{x}_P^{(i)}, \mathbf{v}_P^{(i)})\}_{i=1}^{N_P}$, where $N_P$ denotes the number of atoms in the protein pocket, $\mathbf{x}_P^{(i)} \in \mathbb{R}^3$ and $\mathbf{v}_P^{(i)} \in \mathbb{R}^d$ represent the 3D coordinate and categorical feature of the $i$-th atom. The ligand molecule is represented in a similar way as $\mathcal{M} = \{(\mathbf{x}_M^{(j)}, \mathbf{v}_M^{(j)})\}_{j=1}^{N_M}$, where $N_M$ denotes the number of atoms in the ligand. The size of the ligand $N_M$ can be sampled from an empirical conditional distribution $p(N_M|N_P)$, based on the training dataset that contains ligand-pocket complexes. For brevity, we denote the protein pocket as $\mathbf{p} = [\mathbf{x}_P, \mathbf{v}_P]$, where $\mathbf{x}_P \in \mathbb{R}^{N_P \times 3}$ and $\mathbf{v}_P \in \mathbb{R}^{N_P \times d}$ and $[\cdot, \cdot]$ indicates the concatenation operation. Similarly, the ligand is denoted as $\mathbf{m} = [\mathbf{x}_M, \mathbf{v}_M]$. The SBMO task can be formulated as modeling the conditional distribution $p_\theta(\mathbf{m}|\mathbf{p})$, such that sampled ligand molecules maximize a reward function $r(\mathbf{m}, \mathbf{p})$.

### 3.2 Conditional Equivariant Diffusion Model

Considering SBMO tasks, conditional equivariant diffusion models characterize the conditional distribution $p_\theta(\mathbf{m}|\mathbf{p})$ through a latent variable framework comprising a forward diffusion process and a reverse denoising process. Following the variational diffusion framework used by Kingma et al. (2023); Hoogeboom et al. (2022), the diffusion process progressively corrupts ligand data $\mathbf{m}$ with Gaussian noise over $T$ timesteps. At timestep $t$, the noised data is given as:

$$q(\mathbf{z}_t|\mathbf{m}) = \mathcal{N}(\mathbf{z}_t|\alpha_t \mathbf{m}, \sigma_t^2 \mathbf{I}) \tag{1}$$

where $\alpha_t$ and $\sigma_t$ define the signal-to-noise ratio $\text{SNR}(t) = \alpha_t^2/\sigma_t^2$ at step $t$. The variance-preserving noising process is applied with $\alpha_t = \sqrt{1 - \sigma_t^2}$ and a pre-determined noise schedule introduced in Hoogeboom et al. (2022) is used. The transition distribution of the diffusion process between arbitrary steps $s < t$ can be derived as:

$$q(\mathbf{z}_t|\mathbf{z}_s) = \mathcal{N}(\mathbf{z}_t| \alpha_{t|s}\mathbf{z}_s, \sigma_{t|s}^2 \mathbf{I}) \tag{2}$$

where $\alpha_{t|s} = \alpha_t/\alpha_s$ and $\sigma_{t|s}^2 = \sigma_t^2 - \alpha_{t|s}^2 \sigma_s^2$.

Through Bayes' rule, the true denoising transition conditioned on data $\mathbf{m}$ can be derived in closed form as:

$$q(\mathbf{z}_s|\mathbf{z}_t, \mathbf{m}) = \mathcal{N}(\mathbf{z}_s|\mu_q(\mathbf{z}_t, \mathbf{m}, s, t), \sigma_q^2(s, t)\mathbf{I}) \tag{3}$$

$$\mu_q(\mathbf{z}_t, \mathbf{m}, s, t) = \frac{\alpha_{t|s}\sigma_s^2}{\sigma_t^2}\mathbf{z}_t + \frac{\alpha_s \sigma_{t|s}^2}{\sigma_t^2}\mathbf{m} \ \text{ and } \ \sigma_q(s, t) = \frac{\sigma_{t|s}\sigma_s}{\sigma_t}$$

The reverse denoising process, also known as the generative process, learns to achieve conditional sampling by iteratively refining noisy data via the denoising transition:

$$p_\theta(\mathbf{z}_s|\mathbf{z}_t, \mathbf{p}) = \mathcal{N}(\mathbf{z}_s|\mu_\theta(\mathbf{z}_t, \mathbf{p}, s, t), \sigma_q^2(s, t)\mathbf{I}) \tag{4}$$

By maximizing the variational lower bound for the log-likelihood of the observed data, $\mu_\theta(\mathbf{z}_t, \mathbf{p}, s, t)$ is trained to predict the mean of $q(\mathbf{z}_s|\mathbf{z}_t, \mathbf{m})$. While this optimization can be achieved by having the neural network $\phi_\theta(z_t, \mathbf{p}, t)$ predict $\hat{\mathbf{m}}$, it is often easier to directly predict the noise added in the forward process $\hat{\epsilon} = \phi_\theta(z_t, \mathbf{p}, t)$, then we have $\hat{\mathbf{m}} = \frac{1}{\alpha_t}z_t - \frac{\sigma_t}{\alpha_t}\hat{\epsilon}$.

**Equivariant Denoising Transition.** The denoising transition probability $p_\theta(\mathbf{z}_s|\mathbf{z}_t, \mathbf{p})$ is modeled by EGNN to ensure the equivariance to the group of $O(3)$ transformation $T_g$:

$$p_\theta(T_g(\mathbf{z}_s)|T_g(\mathbf{z}_t, \mathbf{p})) = p_\theta(\mathbf{z}_s|\mathbf{z}_t, \mathbf{p}) \tag{5}$$

Combined with the equivariant prior distribution conditioned on the pocket: $p(T_g(\mathbf{z}_T)|T_g(\mathbf{p})) = p(\mathbf{z}_T|\mathbf{p})$, the generative process acquires a crucial property: the likelihood of the generated ligand given the protein pocket remains invariant under $O(3)$ transformations of the whole ligand-pocket complex:

$$p_\theta(T_g(\mathbf{m})|T_g(\mathbf{p})) = p_\theta(\mathbf{m}|\mathbf{p}) \tag{6}$$

More specifically, $O(3)$-equivariance is achieved by parameterizing $p(\mathbf{z}_T|\mathbf{p})$ and $p_\theta(\mathbf{z}_s|\mathbf{z}_t, \mathbf{p})$ as isotropic Gaussian distributions, where the mean vectors transform equivariantly under $O(3)$ transformations of the ligand-pocket complex.

To ensure equivariance to translations, the diffusion process and generative process are constrained within a linear subspace, where the coordinates of the whole ligand-pocket complex are subtracted by the center of mass of the ligand $1/N_M \sum_i \mathbf{x}_M^i$.

## 4 Our Approach: Denoising Policy for Molecule Optimization

We introduce **De**noising **P**olicy for **P**ocket-**A**ware Molecule Optimization (DEPPA), a novel approach for structure-based molecule optimization that fine-tunes pocket-conditional diffusion models using online reinforcement learning, enabling simultaneous optimization of binding affinity and other desired molecular properties. Concretely, we formulate the reverse denoising process of a pre-trained, pocket-conditional diffusion model as a multi-step Markov Decision Process (MDP) and apply policy optimization to guide the generative process toward desirable molecular outcomes. In this formulation, each denoising step corresponds to a policy action, and the full denoising trajectory constitutes a multi-step decision process conditioned on the protein pocket.

Starting from a pre-trained pocket-conditional diffusion model that defines the conditional distribution $p_\theta(\mathbf{m} \mid \mathbf{p})$, we fine-tune the model parameters such that samples produced by the denoising process increasingly align with the reward function $r(\mathbf{m}, \mathbf{p})$ defined jointly over the ligand $\mathbf{m}$ and the conditional protein pocket $\mathbf{p}$. Formally, the training objective of our approach is to optimize the expected reward of generated ligands sampled from the fine-tuned model:

$$\mathcal{J}_{\text{train}}(\theta; \mathbf{p}) = \mathbb{E}_{\mathbf{m} \sim p_\theta(\mathbf{m}|\mathbf{p})}\big[r(\mathbf{m}, \mathbf{p})\big] \tag{7}$$

The remainder of this section elaborates on the MDP formulation and the denoising policy optimization procedure.

### 4.1 Denoising Procedure as an MDP

We formalize the reverse diffusion dynamics $p_\theta(\mathbf{z}_s \mid \mathbf{z}_t, \mathbf{p})$, which iteratively transforms an initial noisy ligand $\mathbf{z}_T$ into a final generated ligand $\mathbf{z}_0$ conditioned on the protein pocket $\mathbf{p}$, as a Markov Decision Process (MDP). Specifically, we define the MDP as follows:

$$s_t \triangleq (\mathbf{z}_t, t, \mathbf{p}) \qquad a_t \triangleq \mathbf{z}_s \qquad R(s_t, a_t) \triangleq \begin{cases} r(\mathbf{m}, \mathbf{p}), & \text{if } t = 0 \\ 0, & \text{otherwise} \end{cases} \tag{8}$$

$$\pi(a_t \mid s_t) \triangleq p_\theta(\mathbf{z}_s \mid \mathbf{z}_t, \mathbf{p}) \qquad P(s_{t+1} \mid s_t, a_t) \triangleq (\delta_{\mathbf{p}}, \delta_s, \delta_{\mathbf{z}_s})$$

The state $s_t$ consists of the current noisy ligand representation $\mathbf{z}_t$, the diffusion timestep $t$, and the conditional protein pocket $\mathbf{p}$. At each timestep, the policy selects an action $a_t$ defined as $\mathbf{z}_s$, thereby advancing the denoising trajectory. We define the policy $\pi(a_t \mid s_t)$ as the denoising transition probability, modeled as a Gaussian distribution whose mean is parameterized by a neural network $\phi_\theta$. This formulation allows for exact

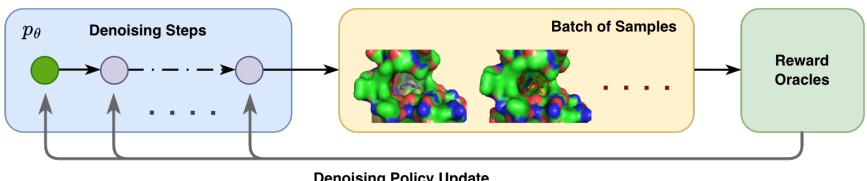

Figure 1: Illustration of one iteration of policy update in DePPA. The green circle stands for $\mathbf{z}_T \sim N(0, I)$. Policy update is performed using a batch of ligand molecules sampled by the denoising policy $p_\theta$ from the previous iteration.

evaluation of the action log-likelihood, enabling gradients of loss functions defined over the log-likelihood to be backpropagated through the network parameters $\theta$. The state transition probability $P(s_{t+1} \mid s_t, a_t)$ is deterministic, where $\delta_y$ denotes the Dirac delta distribution with non-zero density only at y. Note that the transition in the MDP from $s_t$ to $s_{t+1}$ corresponds to a reverse denoising step from time step $t$ to $s$. The reward $r(\mathbf{m}, \mathbf{p})$ is defined with respect to the target protein $\mathbf{p}$ and the generated ligand $\mathbf{m}$, a weighted summation of the property-wise rewards.

## 4.2 Denoising Policy Optimization

To optimize the objective in equation 7, we adopt a variant of Proximal Policy Optimization (PPO) (Schulman et al., 2017) with critic-free advantage estimation to fine-tune the parameters $\theta$ of the denoising policy $\pi(a_t \mid s_t)$. At each policy update iteration, we generate a batch of $N$ denoising trajectories by rolling out the policy from the previous iteration, yielding $N$ sampled ligand molecules. The PPO objective used for policy optimization is defined as follows:

$$L^{\text{CLIP}}(\theta) = \frac{1}{N} \sum_{n=1}^{N} \mathbb{E}_t \left[ \min \left( \omega_t^n(\theta) \hat{A}_t^n, \text{clip}(\omega_t^n(\theta), 1 - \varepsilon, 1 + \varepsilon) \hat{A}_t^n \right) \right] \tag{9}$$

$$\omega_t^n(\theta) = \frac{\pi_\theta(a_t|s_t)}{\pi_{\theta_{old}}(a_t|s_t)} = \frac{p_\theta(\mathbf{z}_s|\mathbf{z}_t, \mathbf{p})}{p_{\theta_{old}}(\mathbf{z}_s|\mathbf{z}_t, \mathbf{p})}$$

Here, $\omega_t^n(\theta)$ denotes the likelihood ratio between the updated denoising policy and its pre-update counterpart at timestep $t$ for trajectory $n$; the advantage $\hat{A}_t^n$ is estimated in a critic-free manner. The clipping operation in $L^{\text{CLIP}}(\theta)$ constrains policy updates to remain within a trust region, thereby preventing excessively large updates that could destabilize training.

**Critic-free Advantage Estimation.** The advantage estimate $\hat{A}_t^n$ at step $t$ for trajectory $n$ is calculated by normalizing the rewards with the group of the $N$ sampled ligands:

$$\hat{A}_t^n = \frac{r(\mathbf{m}, \mathbf{p}) - \mu_r}{\sigma_r} \tag{10}$$

where $\mu_r$ and $\sigma_r$ are the empirical mean and standard deviation of the rewards over the $N$ sampled ligands.

Since we operate in a sparse, outcome-based reward setting—where the reward is only evaluated after a clean ligand is generated—learning a critic function can struggle with credit assignment issues, particularly for early steps in the trajectory. Therefore, instead of using the critic-based advantage estimation applied in standard PPO, we employ a critic-free advantage estimation scheme, inspired by Group Relative Advantage (Shao et al., 2024). The whole trajectory shares the same advantage estimate computed based on the outcome reward.

**Coarser Denoising Scheduler.** Under the framework described in Section 3.2, we can derive the closed form of the denoising transition $p_\theta(\mathbf{z}_s|\mathbf{z}_t, \mathbf{p})$ (see equation 4) for an arbitrary denoising step size $d = t - s$. We propose to employ a larger denoising step size during denoising policy optimization, enabling a principled coarser denoising process which substantially shortens the denoising horizon and reduces the computational cost of sampling. Meanwhile, DePPA uses the same timestep-dependent diffusion schedule as the pretrained model, with the pretrained prior well preserved at the

start of the coarse-step fine-tuning process. Our evaluation results demonstrate that using a larger denoising step size enables efficient yet effective denoising policy optimization. We also empirically study the impact of different denoising step sizes on the performance of DEPPA (see section 5).

The adoption of a coarser denoising scheduler to efficiently navigate chemical space toward high-reward regions can be justified from the following two perspectives:

*(1) Mitigating Error Accumulation:* Fine-tuning pre-trained diffusion models over a long denoising horizon is often unstable due to error accumulation. In the sequential denoising process, minor inaccuracies in early steps amplify across the trajectory, pushing the sample away from the valid data manifold. Increasing the denoising step size mitigates this drift by reducing the total number of denoising transitions, thereby limiting the opportunities for errors to compound.

*(2) More Exploration during the Denoising Process:* The choice of step size directly influences the variance of the denoising transition distribution $p_\theta(\mathbf{z}_s|\mathbf{z}_t, \mathbf{p})$, thereby affecting the exploration-exploitation trade-off during policy optimization. As illustrated in Figure 2, under the variational diffusion framework (Kingma et al., 2023), varying the step size

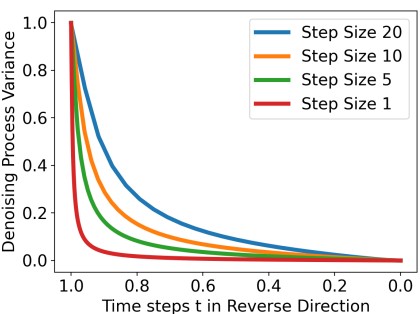

Figure 2: Variance of denoising transitions along the denoising process under different step sizes. Larger denoising step size leads to higher variance of the denoising transition, especially within the middle stages of the denoising process.

yields distinct variance profiles along the denoising process. Essentially, we highlight the natural connection between the denoising step size and the extent of exploration during policy optimization. With a larger step size, the variance of the denoising transitions remains relatively higher, especially within the middle stage of the denoising process. The larger denoising steps allow the RL policy to consider a broader distribution of possible next states and encourage more exploration, preventing the denoising policy from converging prematurely to suboptimal solutions. Nevertheless, we note that excessively large denoising steps can compromise the generation quality, since there can be insufficient exploitation during the later stage of the denoising process.

**Gaussian Rank Transformation of Reward.** In the SBMO setting, the optimization targets multiple heterogeneous molecular properties that differ in numerical scale and distributional shape. To avoid scale imbalance when combining these properties into a weighted-sum reward, we normalize each targeted property using Gaussian rank transformation. Specifically, at each iteration of update, values of each property are first mapped to their batch-wide percentile ranks, and then passed through the inverse Normal CDF, yielding a score that is order-preserving. Notably, it also ensures that the reward weights reflect their intended relative trade-off throughout the training process. This normalization performs a batch-level estimate of how good a molecule is in one property relative to the current policy's sample distribution. As the batch size increases, this statistical estimate becomes less noisy. In the large-batch limit, it converges to a deterministic and monotone property transformation conditioned on the current policy distribution. Consequently, DEPPA's per-iteration update performs a local policy improvement by optimizing a rank-normalized surrogate objective, whose reward is the weighted sum of Gaussian-rank transformed properties computed relative to samples from the current policy. We also conduct ablation study comparing Gaussian rank transformation with z-score and minmax normalizations which are sensitive to outliers, using the same reward weights and training setting (see Table 7 in Appendix C).

**Prediction Thresholding.** While the coarsened denoising schedule reduces the number of iterations and mitigates error accumulation along the trajectory, we additionally apply prediction thresholding to further stabilize the fine-tuning process. Specifically, the predicted clean ligand $\hat{\mathbf{m}} = \frac{1}{\alpha_t}z_t - \frac{\sigma_t}{\alpha_t}\hat{\epsilon}$ is thresholded within the valid range. $\hat{\mathbf{m}}$ is then used to compute the mean of the denoising transition $p_\theta(\mathbf{z}_s|\mathbf{z}_t, \mathbf{p})$. More details regarding the thresholding are discussed in Appendix A.

## 5 Experiments

**Dataset.** Following the same evaluation setting as the previous baselines (Luo et al., 2021; Guan et al., 2023; 2024; Qu et al., 2024; Zhou et al., 2024; Gu et al., 2024; Dorna et al., 2024; Qiu et al., 2024), our experiments are conducted on the CrossDocked2020 dataset (Francoeur et al., 2020), which initially contains 22.5 million protein–ligand complexes. The dataset is refined to 100,000 protein–ligand pairs for training and 100 novel proteins for testing by applying RMSD-based filtering and a 30% sequence-identity split.

**Baselines.** We compare our method to a variety of state-of-the-art baselines that can be categorized into two types. *Generative Methods:* AR (Luo et al., 2021), Pocket2Mol (Peng et al., 2022), FLAG (Zhang et al., 2023) are autoregressive methods. We also include non-autoregressive generative methods including TargetDiff (Guan et al., 2023), DecompDiff (Guan et al., 2024), DiffSBDD (Schneuing et al., 2024), IPDiff (Huang et al., 2024) and MolCRAFT (Qu et al., 2024). *Optimization Methods:* DecompOpt (Zhou et al., 2024) relies on external oracles to evaluate the targeted properties in each round. AliDiff (Gu et al., 2024) and DecompDPO (Cheng et al., 2025) resort to DPO (Wallace et al., 2024) for preference alignment of diffusion models. TAGMol (Dorna et al., 2024) and MolJO (Qiu et al., 2024) leverage gradient-based guidance for optimization.

**Evaluation.** Our evaluation uses metrics as follows: (1) For binding affinity, we report Vina Score, Vina Min and Vina Dock computed by Quick Vina 2 (Alhossary et al., 2015), an accelerated alternative to AutoDock Vina. Specifically, **Vina Score** directly measures the raw binding affinity of the generated ligand pose as-is, **Vina Min** applies a local energy minimization before affinity estimation, **Vina Dock** conducts a re-docking procedure for the generated pose, searching for a pose with optimal binding affinity. **High Affinity**, computed based on Vina Dock, measures the percentage of generated molecules that exhibit stronger binding to each pocket compared to the corresponding reference molecules. (2) Molecular properties, including drug-likeness (**QED**), synthesizability (**SA**), and diversity (**Div**). We also report **Lipinski** as another metric for assessing drug-likeness, which reflects whether a molecule is likely to be an orally active drug. Furthermore, the average size of the generated molecules (**Size**) and the percentage of the generated molecules that are valid and connected (**Complete Rate**) are reported as well. (3) Conformation stability, involving Strain Energy (**SE**) which evaluates the internal energy of generated ligand conformation, Steric Clashes (**Clash**) that calculates the number of clashes within the protein-ligand complex and redocking RMSD (**RMSD**) that measures conformation similarity between the generated and redocked poses. More details regarding the metrics are provided in Appendix B. Consistent with the evaluation protocol used by the other baselines, we sample 100 ligands for each protein pocket in the test set using the fine-tuned model.

**Implementation Details.** DEPPA adopts DiffSBDD (Schneuing et al., 2024) as the pretrained model and fine-tune it for each target pocket, using a reward function defined as a weighted sum of Gaussian rank–transformed scores for Vina Score, QED, SA, and Div. The sensitivity of DEPPA's performance to different reward weights is reported in Appendix C (Table 8). While the denoising horizon of the pretrained model is 500 steps, DEPPA adopts a default denoising step size of 5, resulting in a 5x faster sampling speed relative to the pretrained model. Using larger denoising steps leads to proportional speed-up in sampling. For each iteration of denoising policy optimization, a batch of 32 ligands is sampled, corresponding to 32 denoising trajectories. For each protein pocket, 100 iterations of policy updates are performed, resulting in an average wall-clock overhead of approximately 26 minutes on a single NVIDIA A100 GPU. Regarding the computational cost of the reward oracles, a single evaluation incurs an overhead of approximately 20 ms for Vina Score, 2 ms for QED and less than 1 ms for SA. More implementation details are provided in Appendix A.

### 5.1 Results

In the following, we compare DEPPA with baseline approaches in terms of binding affinity and other molecular properties (Table 1, Table 2), as well as conformational stability (Table 3). We present the stability of DEPPA's optimization in Table 4. We also report ablation studies analyzing the impact of the denoising step sizes in Table 5. We additionally discuss the performance gains achieved by the top-$N$ variant of our approach with detailed results provided in Table 6.

Table 1: Summary of binding affinity and molecular properties for reference molecules and molecules generated by our approach, as well as other generative (Gen.) and optimization (Opt.) baselines. (↑)/(↓) indicates larger / smaller is better. Bold indicates the best performance, while underlined indicates the second best.

| | Methods | Vina Score (↓) Avg. | Vina Score (↓) Med. | Vina Min (↓) Avg. | Vina Min (↓) Med. | Vina Dock (↓) Avg. | Vina Dock (↓) Med. | High Affinity (↑) Avg. | High Affinity (↑) Med. | QED (↑) Avg. | SA (↑) Avg. | Div (↑) Avg. | Size Avg. | Complete (↑) Rate |
|---|---|---|---|---|---|---|---|---|---|---|---|---|---|---|
| | Reference | -6.36 | -6.46 | -6.71 | -6.49 | -7.45 | -7.26 | - | - | 0.48 | 0.73 | - | 22.8 | 100% |
| Gen. | AR | -5.75 | -5.64 | -6.18 | -5.88 | -6.75 | -6.62 | 37.9% | 31.0% | 0.51 | 0.63 | 0.70 | 17.7 | 93.5% |
| | Pocket2Mol | -5.14 | -4.70 | -6.42 | -5.82 | -7.15 | -6.79 | 48.4% | 51.0% | 0.56 | 0.74 | 0.69 | 17.7 | 96.3% |
| | TargetDiff | -5.47 | -6.30 | -6.64 | -6.83 | -7.80 | -7.91 | 58.1% | 59.1% | 0.48 | 0.58 | 0.72 | 24.2 | 90.4% |
| | DecompDiff | -5.67 | -6.04 | -7.04 | -7.09 | -8.39 | -8.43 | 64.4% | 71.0% | 0.45 | 0.61 | 0.68 | 29.4 | 72.0% |
| | DiffSBDD | -1.44 | -4.91 | -4.52 | -5.84 | -7.14 | -7.30 | 35.9% | 30.0% | 0.47 | 0.58 | 0.73 | 24.9 | 84.9% |
| | IPDIFF | -6.41 | -7.01 | -7.45 | -7.48 | -8.57 | -8.51 | 69.5% | 75.5% | 0.52 | 0.59 | 0.74 | 24.1 | 90.1% |
| | MolCRAFT | -6.59 | -7.04 | -7.27 | -7.26 | -7.92 | -8.01 | - | - | 0.50 | 0.69 | 0.72 | 26.8 | 96.7% |
| Opt. | DecompOpt | -5.87 | -6.81 | -7.35 | -7.72 | -8.98 | -9.01 | 73.5% | 93.3% | 0.48 | 0.65 | 0.60 | 32.9 | 71.6% |
| | AliDiff | -7.07 | -7.95 | -8.09 | -8.17 | -8.90 | -8.81 | 73.4% | 81.4% | 0.50 | 0.57 | 0.73 | 24.4 | 92.6% |
| | TAGMol | -7.02 | -7.77 | -7.95 | -8.07 | -8.59 | -8.69 | 69.8% | 76.4% | 0.55 | 0.56 | 0.69 | 24.7 | 92.0% |
| | DecompDPO | -6.13 | -7.54 | -8.30 | -8.57 | **-9.60** | **-9.68** | **85.8%** | **98.5%** | 0.48 | 0.67 | 0.63 | 31.6 | 65.1% |
| | MolJO | -7.52 | -8.02 | -8.33 | -8.34 | -9.05 | -9.13 | - | - | 0.56 | **0.78** | 0.66 | 22.8 | **97.3%** |
| | DᴇPPA | **-8.50** | **-8.63** | **-8.91** | **-8.90** | -9.38 | -9.33 | 79.8% | 88.0% | **0.57** | 0.69 | **0.78** | 21.2 | 95.3% |

**Binding Affinity and Molecule Properties.** As observed in Table 1, DᴇPPA outperforms all baselines in binding affinity with respect to Vina Score and Vina Min, achieving Vina Score of -8.5 kcal/mol, a 13% advantage in Vina Score over the second-best method MolJO and a 33.7% improvement over the reference molecules. Compared to the pretrained model (DiffSBDD) that DᴇPPA fine-tunes, DᴇPPA achieves improvements in Vina Score, Vina Min, and Vina Dock by 490.3%, 97.1% and 31.4%. Moreover, prior work (Qiu et al., 2024; Schneuing et al., 2024) has highlighted a non-negligible correlation between molecular properties and size, with negative Pearson correlation coefficients for binding affinity, drug-likeness and synthesizability. We report an average size of ligand molecules generated by DᴇPPA as 21.2, which is comparable to the average size (22.8) of the reference molecules, indicating that the observed high binding affinity does not arise from size exploitation.

While DecompDPO achieves the best performance in Vina Dock and High Affinity, it generates ligands with a significantly higher average size (31.6) than the reference ligands. Larger ligands provide a greater interaction surface and more opportunities for favorable contacts upon re-docking, which can inflate Vina Dock score and consequently the High Affinity rate, as the latter is computed from Vina Dock score. Meanwhile, DecompDPO's strong Vina Dock performance comes at the expense of drug-likeness and diversity, while also exhibiting a pronounced deficiency in complete rate. We report the High Affinity derived from Vina Score for DᴇPPA as well, showing the improved results (see Table 9 in Appendix C). We highlight that Vina Score serves as a more direct evaluation proxy, as it indicates the raw binding affinity of the generated molecules without post-hoc rearrangement through the re-docking procedure – thereby more credibly reflecting the generative capability of the models.

Regarding molecular properties, DᴇPPA attains the best performance in QED and diversity, thereby highlighting the strong effectiveness of DᴇPPA in multi-objective molecule optimization. The slight lag of DᴇPPA in SA relative to the best-performing method reflects the trade-off between binding affinity and SA identified in prior studies (Qiu et al., 2024; Guan et al., 2024). Nevertheless, DᴇPPA achieves an SA score of 0.69, which remains higher than that of nine out of twelve baselines. In real-world drug discovery scenarios, synthesizability is often used as a coarse screening criterion, whereas binding affinity is regarded as a more critical metric. That said, the minor drawback of DᴇPPA in SA is sufficiently offset by its substantial advantage in binding affinity. Notably, the molecule optimization baselines tend to underperform in diversity compared to the generative baselines. Despite achieving the outstanding performance in binding affinity, DᴇPPA also exhibits the highest diversity score (0.78), outperforming all the baselines and suggesting that it is not collapsing to a narrow set of reward-maximizing templates. Furthermore, as shown in Table 2, DᴇPPA achieves the best results in Lipinski compared to the baselines, surpassing the reference level as well. This provides further evidence for DᴇPPA's superior performance in drug-likeness, given that Lipinski is not directly optimized as a reward signal.

Table 2: Performance in Lipinski results.

| Methods | AR | Pocket2Mol | TargetDiff | IPDiff | DiffSBDD | AliDiff | DEPPA | Reference |
|---|---|---|---|---|---|---|---|---|
| Avg. Lipinski (↑) | 4.75 | 4.88 | 4.51 | 4.52 | 4.56 | 4.48 | **4.96** | 4.27 |

**Conformation Stability.** Recent research (Harris et al., 2023; Qu et al., 2024) has underscored that 3D SBDD approaches may generate molecules outside the true molecular manifold that nonetheless display favorable binding affinity after redocking. Specifically, these ligand molecules fail to capture true interatomic interactions and may even violate biophysical constraints. By relying on post-fixing through the redocking software, the generated poses may undergo significant rearrangements, making credible and accurate pose quality assessments for the generative models increasingly challenging.

We report strain energy, steric clashes, and RMSD between generated and redocked poses to evaluate conformation stability of the generated poses. As demonstrated in Table 3, DEPPA exhibits competitive performance in strain energy compared to other baselines, ranking third among the baselines, which suggests high stability of the generated ligand conformations. Moreover, DEPPA achieves the second-best result in steric clashes and the best performance in RMSD, with 51.8% of ligands closely resembling their post-redocking conformations, indicating outstanding binding-mode consistency. Notably, as shown in Table 1, DEPPA exhibits the smallest changes between Vina Score and Vina Min or Vina Dock, indicating that the raw generated poses capture 3D interatomic interactions within the pocket binding-site both effectively and faithfully. It is also worth emphasizing that the conformation stability metrics are not explicitly optimized as reward signals in DEPPA, yet strong performance is nevertheless observed in this regard. Overall, these results suggest that DEPPA's performance reflects genuine chemical quality improvements rather than exploitation of the scoring pipeline.

Table 3: Evaluation of conformation stability.

| Methods | SE (↓) | | | Clash (↓) | RMSD |
|---|---|---|---|---|---|
| | 25% | 50% | 75% | Avg. | % <2Å(↑) |
| Reference | 34 | 107 | 196 | 5.51 | 34.0 |
| AR | 259 | 595 | 2286 | **4.49** | 31.1 |
| Pocket2Mol | 102 | **189** | **374** | 6.24 | 30.8 |
| FLAG | 143 | 396 | 1164 | 40.83 | 8.2 |
| TargetDiff | 369 | 1243 | 13871 | 10.84 | 29.4 |
| DecompDiff | 115 | 421 | 1424 | 8.16 | 22.7 |
| MolCRAFT | **83** | 195 | 510 | 7.09 | 41.8 |
| DecompDPO | 141 | 323 | 724 | 15.05 | - |
| DEPPA | 108 | 299 | 702 | 5.59 | **51.8** |

**Optimization Stability.** To assess the stability of DEPPA, we report the mean and standard deviation of its average results as evaluated in Table 1 across eight random seeds. As shown in Table 4, this evaluation includes the four optimized reward signals. The low standard deviations indicate the strong stability and robustness of DEPPA 's optimization procedure.

Table 4: DEPPA's mean performance along with standard deviation over eight random seeds, using the default denoising step size of 5.

| | Vina Score (↓) | | QED (↑) | | SA (↑) | | Div (↑) | |
|---|---|---|---|---|---|---|---|---|
| | Mean. | Std. | Mean. | Std. | Mean. | Std. | Mean. | Std. |
| DEPPA | -8.53 | 0.12 | 0.57 | 0.004 | 0.69 | 0.004 | 0.78 | 0.002 |

**Ablation Study.** We evaluate our method with varying denoising step sizes, which directly affects the resolution of RL control in the denoising process and the sampling speed. As shown in Table 5, increasing the step size leads to moderate degradations of performance in binding affinity and synthesizability, while maintaining comparable performance in drug-likeness and diversity. Across various denoising step sizes, the

average sizes of the generated molecules remain similar to each other and comparable to the reference level. Notably, even with a denoising step size of 20, DePPA still exhibits the best average performance in Vina Score, drug-likeness, and diversity compared to the baselines in Table 1, while maintaining competitive results in synthesizability. This observation suggests that, under a constrained computational budget, DePPA with larger step sizes provides a practical and favorable trade-off between sampling speed and optimization quality.

Table 5: Performance of DePPA with different denoising step sizes.

| Methods | Vina Score (↓) | | Vina Min (↓) | | Vina Dock (↓) | | High Affinity (↑) | | QED (↑) | SA (↑) | Div (↑) | Size | Complete (↑) |
|---|---|---|---|---|---|---|---|---|---|---|---|---|---|
| | Avg. | Med. | Avg. | Med. | Avg. | Med. | Avg. | Med. | Avg. | Avg. | Avg. | Avg. | Rate |
| Reference | -6.36 | -6.46 | -6.71 | -6.49 | -7.45 | -7.26 | - | - | 0.48 | 0.73 | - | 22.8 | 100% |
| DePPA (step size=5) | **-8.50** | **-8.63** | **-8.91** | **-8.90** | **-9.38** | **-9.33** | **79.8%** | **88.0%** | 0.57 | **0.69** | **0.78** | 21.2 | **95.3%** |
| DePPA (step size=10) | -7.98 | -8.10 | -8.46 | -8.54 | -9.10 | -9.03 | 77.1% | 85.0% | **0.58** | 0.67 | **0.78** | 21.3 | 95.1% |
| DePPA (step size=20) | -7.59 | -7.68 | -8.01 | -7.96 | -8.62 | -8.54 | 73.9% | 83.0% | **0.58** | 0.65 | **0.78** | 21.1 | 91.7% |

**Top-$N$ Variant of DePPA.** We also report a variant of our approach (DePPA†), which selects top-$N$ ligands explored during the RL fine-tuning for each protein pocket instead of sampling from the fine-tuned denoising policy. As reported in Table 6, DePPA† yields an even more pronounced advantage in binding affinity relative to DePPA, achieving Vina Score improvements of 59.0% for N=10 and 39.8% for N=100. High Affinity ratios for N=10 and N=100 even surpass 96%. These improvements in binding affinity are accompanied by an increase in the average size of generated ligands, as well as moderate degradation in QED, SA, and diversity, which is expected as the binding affinity, QED and SA are correlated to the size of molecules (Qiu et al., 2024). With N=100 or N=10, DePPA† significantly outperforms all methods reported in Table 1 in binding affinity.

Table 6: Comparison of top-$N$ variants of DePPA with N=100 and N=10.

| Methods | Vina Score (↓) | | Vina Min (↓) | | Vina Dock (↓) | | High Affinity (↑) | | QED (↑) | SA (↑) | Div (↑) | Size |
|---|---|---|---|---|---|---|---|---|---|---|---|---|
| | Avg. | Med. | Avg. | Med. | Avg. | Med. | Avg. | Med. | Avg. | Avg. | Avg. | Avg. |
| Reference | -6.36 | -6.46 | -6.71 | -6.49 | -7.45 | -7.26 | - | - | 0.48 | 0.73 | - | 22.8 |
| DePPA | -8.50 | -8.63 | -8.91 | -8.90 | -9.38 | -9.33 | 79.8% | 88.0% | **0.57** | **0.69** | **0.78** | 21.2 |
| DePPA† (N=100) | -11.88 | -11.75 | -12.18 | -12.08 | -12.37 | -12.28 | 96.2% | **100%** | 0.55 | 0.64 | 0.64 | 28.5 |
| DePPA† (N=10) | **-13.52** | **-13.46** | **-13.85** | **-13.76** | **-13.92** | **-13.88** | **96.6%** | **100%** | 0.51 | 0.60 | 0.60 | 32.4 |

## 6 Conclusion

This paper introduced DePPA, a novel framework for structure-based molecule optimization that leverages online reinforcement learning to fine-tune a pre-trained, pocket-aware diffusion model. DePPA formulates the reverse denoising process as a sequential decision-making problem and applies denoising policy optimization, incorporating critic-free advantage estimation, a coarser denoising scheduler, Gaussian rank transformation of rewards and prediction thresholding. This enables lightweight, efficient, and effective denoising policy optimization under sparse, outcome-based reward signals. Moreover, we reveal the natural connection between the denoising step size and exploration extent during the RL policy optimization.

Experimental results on the CrossDocked2020 benchmark demonstrate that DePPA consistently outperforms state-of-the-art baselines across multiple objectives. In particular, DePPA achieves a substantial advantage in binding affinity, drug-likeness, and molecular diversity, while maintaining strongly competitive performance in synthesizability. Notably, to the best of our knowledge, DePPA is the first method to achieve a Vina Score of $-8.5$ kcal/mol while generating ligands with sizes comparable to reference molecules. In addition, we show that ligands generated by DePPA exhibit strong conformational stability, as evidenced by outstanding performance in RMSD, steric clash, and strain energy metrics. Furthermore, the strong stability of the fine-tuning process and the impact of varying denoising step sizes are empirically reported. We also show that the simple top-N filtering can further enhance binding affinity, highlighting DePPA as a strong foundation for downstream selection and refinement pipelines. Although our evaluation of DePPA em-

ploys DiffSBDD as the pretrained model, the significant improvements across all metrics indicate promising potential of applying DePPA to other diffusion-based molecular generators as future work.

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

## A    More Implementation Details

**Reward Processing.**    In DEPPA, we feed raw continuous values of Vina score and molecule properties to the Gaussian rank transformation, resulting in a zero-centered distribution that resembles a normal distribution. For Vina Score, the evaluated values are reversed before the Gaussian rank transformation. The transformed values are then combined into the weighted-sum reward signal for each generated ligand as: $\mathcal{R} = 0.2*\text{QED} + 0.2*\text{SA} + 0.5*\text{Vina Score} + 0.1*\text{Diversity}$. Notably, Diversity is optimized by minimizing intra-batch similarity, quantified as the mean Tanimoto similarity computed from Morgan fingerprints between each generated ligand and the remaining ligands in the same batch. For invalid molecules, we assign a penalty reward defined as $(\mu_B - 3\sigma_B)$, where $\mu_B$ and $\sigma_B$ are the mean and standard deviation of the weighted-sum rewards associated with valid ligands in the same batch. This formulation maintains a consistent relative penalty strength across each batch of generated ligands.

**Prediction Thresholding.**    The atom-type features are scaled by a factor of 0.25, as empirically validated in previous work (Hoogeboom et al., 2022). Accordingly, the atom-type features of the predicted clean ligand $\hat{\mathbf{m}}$ are thresholded to the range $[0, 0.25]$, while the predicted coordinates are left unchanged.

**Top-$N$ Variant of DePPA.**    Among all ligands generated during the full RL fine-tuning process, the top-$N$ ligands are ranked and filtered based on a weighted z-score sum: $z = 5\,norm(|\text{Vina Score}|) + norm(\text{QED}) + 1.5\,norm(\text{SA})$, following the setting applied in Qiu et al. (2024).

**RL Fine-tuning Details.**    DEPPA employs DiffSBDD (Schneuing et al., 2024) as the pretrained model to fine-tune; the noise schedule and the architecture of the denoising policy in DEPPA remain the same as in DiffSBDD. During policy updates, we employ AdamW optimizer with a learning rate of $1e-5$ and a weight decay of $1e-4$. The clipping range for policy update is 0.2.

## B    More Details on the Metrics

Beyond the commonly reported metrics in prior work—namely binding affinity scores (Vina Score, Vina Min, Vina Dock), as well as QED and SA—we further elaborate on the other evaluation metrics below.

- **Diversity** measures pairwise Tanimoto similarity based on Morgan fingerprints among the generated molecules; this metric reflects the diversity of the generated ligands given the target binding site.

- **Strain Energy** quantifies the internal energy of generated ligand poses and serves as an indicator of their conformational stability.

- **Steric Clash** calculates the number of steric clashes between the generated ligand and the protein surface, where a clash is defined as ligand and protein atoms being closer than a specified distance threshold.

- **Redocking RMSD** computes symmetry-corrected Root-Mean-Square Deviation between the generated ligand and Vina redocked poses, measuring conformation similarity. Redocking RMSD smaller than 2Å indicates consistency between the generated and redocked binding-modes, reflecting a strong capability of a generative model at capturing the 3D interaction within the protein-ligand complex.

## C    More Experimental Results

**Ablation of reward normalization.**    We also conduct ablation study to compare Gaussian rank transformation to z-score and minmax reward normalization, using the same reward weights and training setting. As presented in Table 7, DePPA with Gaussian rank transformation outperforms z-score and minmax normalization in Vina Score while maintaining comparable performance in QED, SA and Div.

Table 7: DEPPA's performance in mean and standard deviation across eight random seeds with different reward normalizations, using the default denoising step size of 5.

| Reward Normalization | Vina Score (↓) | | QED (↑) | | SA (↑) | | Div (↑) | |
|---|---|---|---|---|---|---|---|---|
| | Mean. | Std. | Mean. | Std. | Mean. | Std. | Mean. | Std. |
| Gaussian rank (default) | -8.53 | 0.12 | 0.57 | 0.004 | 0.69 | 0.004 | 0.78 | 0.002 |
| z-score | -8.18 | 0.18 | 0.57 | 0.003 | 0.69 | 0.003 | 0.78 | 0.003 |
| minmax | -7.77 | 0.21 | 0.57 | 0.003 | 0.70 | 0.004 | 0.78 | 0.003 |

**Sensitivity to reward weights.** We also investigate the sensitivity of DePPA's performance to changes in the reward weights under the same training setting. As reported in Table 8, the performance varies in line with the intended effects of the corresponding adjustments of the reward weights. Specifically, increasing the weight of Vina Score while decreasing the weights of QED and SA leads to increased performance in Vina Score and reduced performance in QED and SA, and vice versa. While the weight of Diversity remains unchanged, assigning a higher weight to the Vina Score tends to result in mildly decreased diversity.

Table 8: DEPPA's performance in mean and standard deviation across eight random seeds under different reward weights in the order of (Vina Score - QED - SA - Div). The default denoising step size of 5 is used.

| Reward Weights | Vina Score (↓) | | QED (↑) | | SA (↑) | | Div (↑) | |
|---|---|---|---|---|---|---|---|---|
| | Mean. | Std. | Mean. | Std. | Mean. | Std. | Mean. | Std. |
| 0.3-0.3-0.3-0.1 | -5.61 | 0.50 | 0.60 | 0.003 | 0.74 | 0.003 | 0.80 | 0.003 |
| 0.5-0.2-0.2-0.1 (default) | -8.53 | 0.12 | 0.57 | 0.004 | 0.69 | 0.004 | 0.78 | 0.002 |
| 0.7-0.1-0.1-0.1 | -9.01 | 0.08 | 0.54 | 0.003 | 0.61 | 0.004 | 0.76 | 0.003 |

**High Affinity based on Vina Score.** We also report High Affinity computed based on Vina Score for the variants of DEPPA, as presented in Table 9. High Affinity derived from Vina Score is improved over the one based on Vina Dock for all variants of DEPPA.

Table 9: Comparison of High Affinity computed based on Vina Dock and Vina Score (labeled with an asterisk).

| DEPPA Variants | High Affinity (↑) | | High Affinity* (↑) | |
|---|---|---|---|---|
| | Avg. | Med. | Avg. | Med. |
| DEPPA (step size=5) | 79.8% | 88.0% | 83.1% | 93.0% |
| DEPPA (step size=10) | 77.1% | 85.0% | 81.1% | 92.0% |
| DEPPA (step size=20) | 73.9% | 83.0% | 78.2% | 89.0% |
| DEPPA† (N=100) | 96.2% | 100% | 98.1% | 100% |
| DEPPA† (N=10) | 96.6% | 100% | 98.6% | 100% |

