# OpenReview forum: "Fine-tuning Pocket-Aware Diffusion Models via Denoising Policy Optimization"
_TMLR — Under review for TMLR_

### Review · Reviewer_7PeG · 2026-06-07

**Summary Of Contributions:**

This paper introduces DEPPA (Denoising Policy for Pocket-Aware Molecule Optimization), an online reinforcement learning (RL) framework designed for structure-based molecule optimization (SBMO). Built upon a pre-trained pocket-conditional 3D diffusion model, DEPPA formulates the iterative reverse denoising process as a multi-step MDP. It directly leverages continuous feedback from property oracles as outcome-based rewards to fine-tune the generative policy. To bypass the challenges of long-horizon credit assignment and high computational costs, the authors incorporate several stability and efficiency enhancements. Evaluated on the CrossDocked2020 benchmark, DEPPA significantly improves raw binding affinity while achieving top-tier or highly competitive multi-property and conformational stability metrics.

Strengths
- Strong structural and chemical performance in empirical experiments
- Robust scale invariance: The use of Gaussian rank transformation effectively solves the problem of dominant large-scale properties overwhelming multi-objective rewards.
- The paper formulates the reverse denoising process as a multi-step MDP and adapts policy optimization to pocket-aware molecular diffusion models in a principled manner. The formulation is technically sound and clearly presented.

Weaknesses
- Since Vina-based scores are used both as optimization rewards and as primary evaluation metrics, it remains unclear whether improvements generalize to alternative docking engines or external affinity estimators.
- The reward function uses manually chosen weights across multiple objectives, but the sensitivity of performance to these choices is not investigated.
- It is not entirely clear whether RL fine-tuning is performed independently for each target pocket or whether a single optimized model is used across targets. This distinction has important implications for both computational cost and practical deployment.
- The paper argues that DPO-based approaches are limited by preference datasets, but DePPA benefits from online reward optimization. Since the two paradigms operate under different supervision settings, a more careful discussion of the comparison would strengthen the paper.

**Audience:**

Yes

**Audience Explanation:**

The paper studies an important problem in structure-based molecule optimization and combines ideas from diffusion models and reinforcement learning, two areas of significant interest to the TMLR community. The proposed approach demonstrates strong empirical performance and provides practical insights into RL-based fine-tuning of diffusion models. Researchers working on generative modeling, reinforcement learning, scientific machine learning, and AI for drug discovery would likely find the findings relevant and useful, even if some of the methodological components build upon existing techniques.

**Claims And Evidence:**

Yes

**Claims Explanation:**

The paper provides clear and generally convincing empirical evidence supporting its main claims. The proposed method is evaluated on a standard benchmark against a broad range of competitive baselines and demonstrates consistent improvements in binding affinity while maintaining strong performance on molecular property metrics. Additional evaluations on conformation stability and optimization robustness further strengthen the empirical support. While the paper would benefit from more extensive ablations and stronger analysis of certain design choices, the presented evidence is sufficient to support the primary claims of the work.

**Requested Changes:**

- The paper reports both Avg. and Med. for several metrics, but it is unclear which serves as the primary evaluation criterion. For example, on Page 10, the authors state that "even with a denoising step size of 20, DePPA still exhibits the best performance in Vina Score." This is true for the average Vina Score, but not for the median, where MolJO achieves a better result (-8.02 vs. -7.68). The authors should clarify which statistic is used for comparison and ensure the conclusions are consistent with it.
- Equation 8 misses delta definitions and r(m, p). The authors must completely re-typeset Section 4.1 using unified, formal LaTeX blocks to ensure mathematical clarity, correct notation, and standard terminal punctuation.
- In Section 4.1, the MDP transition probability is defined forward in operational sequence space as $P(s_{t+1}|s_t, a_t)$, whereas the diffusion process itself steps backwards in chronological time from T down to 0. To prevent this notation from looking like a contradiction to readers tracking index variables, the authors should add a brief sentence or explicit footnote confirming the notation usages of denoting steps and policy time step.

---

> ### Author Response · Authors · 2026-07-16
> **Official Response by Authors**
>
> We thank the reviewer for the constructive and insightful review. In the following, we address each point raised and indicate the corresponding changes in the revised manuscript (marked in blue).
>
> **Q1: Generalization to alternative docking engines.**
>
> Ans: DePPA optimizes binding affinity evaluated by QuickVina2, which is also used by the pretrained model DiffSBDD. To demonstrate the generalization performance, we also report the binding affinity evaluation using Autodock Vina, another docking engine adopted by several baselines including TargetDiff, DecompDiff, AliDiff and MolJO. As shown in the table below, we re-evaluate the same samples reported in Table 1 of the original manuscript using Autodock Vina, and the results show comparable performance across the two docking engines.
>
> |**Docking Engine**|**Vina** |   **Score**   | **Vina**  |   **Min** |   **Vina** |  **Dock**  |
> |:-----------------|:---------:|:---------:|:---------:|:---------:|:---------:|:---------:|
> |                  | Avg      | Med      | Avg      | Med      | Avg      | Med      |
> | QuickVina2       | -8.50    | -8.63     | -8.91     | -8.90    | -9.38     | -9.33    |
> | Autodock Vina    | -8.50    | -8.62     | -8.85     | -8.83    | -9.38     | -9.33    |
>
> **Q2: Sensitivity of performance to the weights across multiple objectives.**
>
> Ans: The default reward weights are chosen based on the following considerations. First, binding affinity is the primary objective in SBMO and is therefore given the greatest emphasis. Second, while higher SA and QED scores are generally desirable, these properties are mainly used as screening metrics in real-world drug discovery, and molecules are typically considered acceptable as long as they remain within reasonable ranges. Finally, diversity is mildly weighted to promote exploration and prevent convergence to local optima.
>
> We also conduct additional experiments to investigate the sensitivity of DePPA’s performance to different reward weights. As shown in the table below, the resulting performance varies in line with the intended effects of the corresponding weight adjustments. Specifically, increasing the weight of Vina Score while decreasing the weights of QED and SA leads to increased performance in Vina Score and reduced performance in QED and SA, and vice versa. While the weight of Diversity remains unchanged, assigning a higher weight to the Vina Score tends to mildly reduce diversity. We’ve also added these results and the related discussion to Appendix C.
>
> |        **Weights**            |**Vina**|      | **QED**  |          |   **SA** |          |  **Div** |          |
> |:----------------|---------:|---------:|---------:|---------:|---------:|---------:|---------:|---------:|
> |       Vina-QED-SA-Div    | Mean     | Std      | Mean     | Std      | Mean     | Std      | Mean     | Std      |
> | 0.3-0.3-0.3-0.1 | -5.61    | 0.50     | 0.60     | 0.003    | 0.74     | 0.003    | 0.80     | 0.003   |
> | 0.5-0.2-0.2-0.1 (default) | -8.53    | 0.12     | 0.57     | 0.004    | 0.69     | 0.004    | 0.78     | 0.002    |
> | 0.7-0.1-0.1-0.1  | -9.01    | 0.08     | 0.54     | 0.003    | 0.61     | 0.004    | 0.76     | 0.003    |
>
> **Q3: RL fine-tuning for each target pocket or one optimized model across targets.**
>
> Ans: The RL fine-tuning of DePPA is performed separately for each target pocket, using the same set of hyperparameters. The implementation details described in Section 5 and Appendix A, including the training budget, are based on this target-specific optimization setting.
>
> **Q4: Difference in supervision setting between DPO-based approaches and DePPA.**
>
> Ans: We appreciate the insightful comment and agree that the online characteristic of DePPA should be more explicitly pointed out in the comparison between the two paradigms. We have correspondingly updated Introduction and Related Work of the manuscript, clarifying that the referenced DPO-based approaches depend on the fixed and offline preference dataset, whereas DePPA employs an online RL setup.
>
> **Q5: More precise description of performance with denoising step size 20.**
>
> Ans: We thank the reviewer for pointing this out. We’ve made a revision to clearly specify that DePPA, even with denoising step size 20,  still outperforms baselines in the average performance in Vina Score (see Section 5, Ablation Study).
>
> **Q6: Improve mathematical clarity and notation in section 4.1.**
>
> Ans: We have updated the corresponding description in section 4.1 to enhance the clarification regarding the definition of the Dirac delta distribution and $r(m,p)$. The latex formatting of Equation 8 has also been adjusted for more clear readability.
>
> **Q7: Augment description of the MDP transition probability.**
>
> Ans: We thank the reviewer for pointing out this potential ambiguity. We've revised the description of the MDP transition probability in section 4.1 to avoid the potential confusion between the denoising time steps and the policy time steps.

---

### Review · Reviewer_mns6 · 2026-06-08

**Summary Of Contributions:**

The proposed DePPA formulates the reverse denoising process of a pretrained pocket-aware diffusion model as a Markov Decision Process (MDP) and fine-tunes it using DDPO/PPO-style reinforcement learning, thereby steering molecular generation toward higher binding affinity and drug-like properties for a given protein pocket. To enable stable optimization in the structure-based molecule optimization setting, the authors further introduces a coarse denoising schedule, critic-free advantage estimation, and rank-based reward normalization.

**Audience:**

Yes

**Audience Explanation:**

The problem setting (generating ligand molecules) is a well-studied theme mainly motivated from drug design, and thus, combining diffusion models with PPO would be sufficiently attractive for TMLR's audience.

**Claims And Evidence:**

Yes

**Claims Explanation:**

The results suggest that the proposed framework sufficiently high performance for structure-based drug design task (e.g., wrt binding affinity).

**Requested Changes:**

I have a few technical questions that I hope the authors clarify. In particular, I do not fully understand an advantage of the rank-based transformation as described below.

The discussion comparing the proposed DePPA with DecompDPO could be further clarified. The manuscript highlights two key differences: (i) DecompDPO relies on pairwise preference, and (ii) its performance may be limited by the need to construct a high-quality preference dataset in advance, leading to the claim that the optimization frontier can be bounded by the support and diversity of the preference data.
- (i) However, I found the distinction somewhat less clear than suggested. Regarding (i), DePPA ultimately applies a rank-based transformation to rewards within each batch before policy optimization. Since this transformation essentially discards the absolute reward information and converts to the relative ordering, it is not immediately obvious to me whether the information utilized by DePPA is fundamentally richer than that available through pairwise preference-based objectives.
- (ii) Regarding (ii), if reward oracles are available, it seems straightforward to convert reward evaluations into pairwise preference relations. Furthermore, one could imagine generating new samples, evaluating them with the oracle, and continuously constructing additional preference instances during training, i.e., an online preference-learning setting rather than relying on a fixed offline preference dataset. From this perspective, it is not entirely clear why the use of pairwise preferences per se would necessarily imply a bounded optimization frontier. It would be helpful if the authors could more explicitly explain whether the claimed limitation arises from the pairwise preference-learning objective itself.

I would appreciate further discussion regarding the theoretical motivation for the Gaussian rank transformation being performed at the 'batch level'. Since the transformation is computed using the relative ranking of samples within a batch, the same molecule could receive different transformed reward depending on the composition of the batch in which it appears. In other words, the objective function can be different for each batch by which the connection between the original definition of the optimization problem and the proposed method actually optimizes becomes unclear. Further, empirical evidence of the rank transformation (performance gain against the proposed model without rank transformation) is not provided.

The denoising step size appears to be a key component of the proposed method, and Table 5 shows that performance varies noticeably with this choice. However, it is unclear how the default step size of 5 was selected. If the authors selected it based on the result of Table 5, the results of Table 1 becomes somewhat unfairly biased.

The reward function uses a fixed weighted combination of Vina, QED, SA, and Diversity (0.5, 0.2, 0.2, and 0.1, respectively). Could the authors clarify how these particular weights were chosen? In addition, it would be helpful to discuss whether these values are motivated by domain knowledge, tuned empirically, or selected through some validation procedure, and what justification exists for this specific trade-off among the objectives. Further, to be honest, I feel the use of the term 'multi-objective optimization' in the Introduction may be somewhat misleading because DePPA doesn't have particular methodological novelty of dealing with multiple criteria (just using a fixed scalarized score).

In my understanding, the basic framework of the proposed method can be seen as a combination of DDPO and DiffSBDD. In Introduction, more elaborate descriptions about which components are inherited from DDPO and DiffSBDD should have been revealed, though the authors briefly discussed them, to clarify the positioning of the proposed method relative to those prior studies.

---

> ### Author Response · Authors · 2026-07-16
> **Official Response by Authors**
>
> We thank the reviewer for the thoughtful and detailed feedback. In the following, we respond to each comment in turn and outline the corresponding revisions made to the manuscript, which are marked in blue.
>
> **Q1: Information utilised by DePPA’s objective and pairwise preference-based objective.**
>
> Ans: The pairwise preference-based objective gets saturated once the policy ranks a pair of positive and negative samples correctly, regardless of how different they are in the target metrics. Moreover, a moderately good molecule and a top-ranked molecule can both be labeled as positive samples, leaving their relative advantage in the targeted metrics indistinguishable to the objective. In contrast, for an update iteration of DePPA, the batch-level Gaussian rank transformation–while likewise discarding the absolute reward magnitudes–preserves the batch-wide ordering, enabling a graded training signal that is richer than binary pairwise labels. The batch-relative percentile ranking yields an empirical estimate of the current sampling policy’s population reward CDF.
>
> **Q2: Discussion regarding the online preference-learning setting.**
>
> Ans: We thank the reviewer for this observation and agree that the pairwise preference-learning objective is not inherently limited to a bounded optimization frontier; rather, this frontier depends on how the preference data is obtained. In the offline setting used by our DPO-based baselines, the DPO loss is defined over the support of a fixed pre-collected preference dataset, so the achievable optimization frontier is bounded by the dataset's quality and coverage. This limitation can be alleviated in an online preference-learning setting: given access to reward oracles, one can sample from the continually updated policy to construct new preference data on the fly, making the support of the training data dynamic as optimization proceeds. However, doing so reintroduces the repeated sampling and oracle evaluation that the typical online RL loop requires—precisely the overhead DPO was designed to eliminate—while still relying on a binary preference signal that is a lossy compression of the original scalar rewards. We've revised the Introduction to make the claim more precise.
>
> **Q3: Clarification regarding the motivation of Gaussian rank transformation and the corresponding ablation study.**
>
> Ans: DePPA targets multiple heterogeneous molecular properties with different numerical scales and distributional shapes. Across iterations, Gaussian rank transformation maps each molecular property to standard Gaussian shape before forming the weighted combination, which addresses scale imbalance and outliers. Notably, it ensures that the reward weights can reflect their intended relative priority throughout the training process.
>
> During a single iteration of update, a batch of molecules is not arbitrary but sampled by the current policy. The batch-level Gaussian rank transformation-an order-preserving normalization–then estimates how good a molecule is relative to the current policy’s sample distribution, not relative to a fixed absolute reward scale. By increasing the batch size, this empirical estimate becomes less noisy. In the large-batch limit, it converges to a deterministic and monotone reward transformation conditioned on the current policy distribution. Consequently, DePPA’s per-iteration update performs a local policy improvement step: it optimizes a rank-normalized surrogate objective whose reward is the weighted sum of Gaussian-rank transformed properties computed relative to samples from the current policy. As the policy improves, the empirical reference distribution also shifts such that a molecule that was top ranked may become average later, which is expected with an objective that reflects relative improvement. We’ve revised the discussion regarding Gaussian rank transformation in Section 4.2.
>
> We further agree that empirical evidence is needed, and we’ve conducted additional experiments to compare Gaussian rank transformation to z-score and minmax reward normalization. As presented in the table below, DePPA with Gaussian rank transformation outperforms z-score and minmax normalization in Vina Score while maintaining comparable performance in QED, SA and Div. And we’ve added these ablation results to Appendix C.
>
>
> |        **Normalization**            |**Vina**|      | **QED**  |          |   **SA** |          |  **Div** |          |
> |:----------------|---------:|---------:|---------:|---------:|---------:|---------:|---------:|---------:|
> |           | Mean     | Std      | Mean     | Std      | Mean     | Std      | Mean     | Std      |
> | Gaussian rank | -8.53    | 0.12     | 0.57     | 0.004    | 0.69     | 0.004    | 0.78     | 0.002    |
> | z-score                | -8.18    | 0.18     | 0.57     | 0.003    | 0.69     | 0.003    | 0.78     | 0.003    |
> | minmax                 | -7.77    | 0.21     | 0.57     | 0.003    | 0.70     | 0.004    | 0.78     | 0.003    |

---

> ### Author Response · Authors · 2026-07-16
> **Official Response by Authors**
>
> **Q4: Discussion regarding the selection of the default denoising step size.**
>
> Ans: The denoising step size is a DePPA-specific hyperparameter that is tied to the diffusion framework we choose. Table 5 is provided as a sensitivity analysis of this hyperparameter, not as post-hoc tuning for Table 1. It shows a clear performance-cost trade-off: as the step size increases from 5 to 10 to 20, sampling becomes cheaper, but performance decreases monotonically, especially for binding affinity. Hence, step size 5 is used as the default because it gives the best overall performance across the reported metrics while remaining 5x faster than the original sampler. Importantly, Table 5 shows that even at step size 20, DePPA maintains a highly competitive performance compared to the baselines, suggesting a favorable trade-off for practitioners sensitive to training cost.
>
> **Q5: Clarification of how the reward weights are chosen and presentation of sensitivity analysis.**
>
> Ans: The reward weights are chosen based on the following motivations. First, binding affinity is the primary objective in SBMO and is therefore given the greatest emphasis. Second, while higher SA and QED scores are generally desirable, these properties are mainly used as screening metrics in real-world drug discovery, and molecules are typically considered acceptable as long as they remain within reasonable ranges. Finally, diversity is mildly weighted to promote exploration and prevent convergence to local optima. We have conducted additional experiments with different combinations of reward weights. As presented in the table below, the performance changes in a predictable manner according to the weights adjustments. We’ve added these results to Appendix C in the manuscript.
>
> |        **Weights**            |**Vina**|      | **QED**  |          |   **SA** |          |  **Div** |          |
> |:----------------|---------:|---------:|---------:|---------:|---------:|---------:|---------:|---------:|
> |       Vina-QED-SA-Div    | Mean     | Std      | Mean     | Std      | Mean     | Std      | Mean     | Std      |
> | 0.3-0.3-0.3-0.1 | -5.61    | 0.50     | 0.60     | 0.003    | 0.74     | 0.003    | 0.80     | 0.003   |
> | 0.5-0.2-0.2-0.1 (default) | -8.53    | 0.12     | 0.57     | 0.004    | 0.69     | 0.004    | 0.78     | 0.002    |
> | 0.7-0.1-0.1-0.1  | -9.01    | 0.08     | 0.54     | 0.003    | 0.61     | 0.004    | 0.76     | 0.003    |
>
> **Q6: Positioning of the proposed method relative to DDPO and DiffSBDD.**
>
> Ans: The reviewer is correct that DePPA draws inspiration from DDPO [1], specifically in formulating the reverse denoising process as a multi-step Markov Decision Process. Beyond this high-level formulation, however, all four core components introduced in DePPA are absent from DDPO. DiffSBDD [2], on the other hand, is the pretrained model that DePPA fine-tunes. It is built upon the variational diffusion framework proposed by Kingma et al. [3], which we retain in DePPA. Preserving this framework is essential for establishing the connection between the coarse scheduler and the exploration behavior of the denoising policy, an insight that we consider a key methodological contribution. To better clarify these distinctions, we have revised the Introduction of the manuscript accordingly.
>
> [1] Black, Kevin, et al. "Training diffusion models with reinforcement learning." International Conference on Learning Representations. Vol. 2024. 2024.
>
> [2] Schneuing, Arne, et al. "Structure-based drug design with equivariant diffusion models." Nature Computational Science 4.12 (2024): 899-909.
>
> [3] Kingma, Diederik, et al. "Variational diffusion models." Advances in neural information processing systems 34 (2021): 21696-21707.